# Fragment-Based Drug Discovery against Mycobacteria: The Success and Challenges

**DOI:** 10.3390/ijms231810669

**Published:** 2022-09-14

**Authors:** Namdev S. Togre, Ana M. Vargas, Gunapati Bhargavi, Mohan Krishna Mallakuntla, Sangeeta Tiwari

**Affiliations:** Department of Biological Sciences & Border Biomedical Research Centre, University of Texas at El Paso, El Paso, TX 79968, USA

**Keywords:** NTM, mycobacteria, FBDD, drug discovery, NTM drug discovery, Mtb drug discovery

## Abstract

The emergence of drug-resistant mycobacteria, including Mycobacterium tuberculosis (Mtb) and non-tuberculous mycobacteria (NTM), poses an increasing global threat that urgently demands the development of new potent anti-mycobacterial drugs. One of the approaches toward the identification of new drugs is fragment-based drug discovery (FBDD), which is the most ingenious among other drug discovery models, such as structure-based drug design (SBDD) and high-throughput screening. Specialized techniques, such as X-ray crystallography, nuclear magnetic resonance spectroscopy, and many others, are part of the drug discovery approach to combat the Mtb and NTM global menaces. Moreover, the primary drawbacks of traditional methods, such as the limited measurement of biomolecular toxicity and uncertain bioavailability evaluation, are successfully overcome by the FBDD approach. The current review focuses on the recognition of fragment-based drug discovery as a popular approach using virtual, computational, and biophysical methods to identify potent fragment molecules. FBDD focuses on designing optimal inhibitors against potential therapeutic targets of NTM and Mtb (PurC, ArgB, MmpL3, and TrmD). Additionally, we have elaborated on the challenges associated with the FBDD approach in the identification and development of novel compounds. Insights into the applications and overcoming the challenges of FBDD approaches will aid in the identification of potential therapeutic compounds to treat drug-sensitive and drug-resistant NTMs and Mtb infections.

## 1. Introduction

The World Health Organization (WHO) has reported global tuberculosis (TB) with an estimated 10 million active new cases, 1.5 million deaths, and 1.7 billion latent infections in 2020 [1]. Although the effective treatment of TB has been available for over four decades, the occurrence of drug resistance (DR) and multi-drug resistance (MDR) is a major challenge during the treatment course. According to the recent WHO report, although the total number of DR cases decreased by 22% in 2019–2020, the existing drug regimens are ineffective in combating the disease progression [1,2]. Recent studies identified fragment-based drug discovery (FBDD) as a key approach in the field of drug discovery to find new targets through high-throughput screening (HTS) techniques [3,4]. The identification of even smaller molecules, the druggability of biological targets, and alternate inhibition sites on established drugs are key to the FBDD’s success [5,6]. Based on this approach, more than 40 new compounds have been identified and are currently in clinical trials [7,8,9]. Figure 1 shows the basic approach used to identify new compounds in FBDD.

This FBDD approach has evolved steadily over the last decade. The application of fragment library screening has been successfully applied to a wide range of infectious diseases, including, but not limited to, *Leishmania*, flaviviruses, non-tuberculosis mycobacteria (NTM), and Mtb [10]. The discovery of the phosphoribosylaminoimidazole succinocarboxamide synthetase enzyme (PurC) as a potential inhibitor target in both *M. abscessus* and Mtb is also a significant breakthrough in FBDD [11]. It is crucial to understand how beneficial the FBDD approach is in developing new molecules with possible potential therapeutic activity against NTM and Mtb infections. Therefore, in the present review, we summarize strategies, applications, and challenges in drug discovery using the FBDD approach.

## 2. Fragment-Based Screening Methods

Several studies have already explained the efficiency of FBDD and its coalition with the computational approach and biophysical techniques in the identification of potent inhibitors. This section will particularly focus on the widely used biophysical techniques in FBDD used for the identification of anti-Mtb and anti-NTM molecules.

### 2.1. Virtual Screening and Computational Methods

Virtual screening (VS), a computer-based technique used to predict a compound’s ability to bind target protein, has been extensively reviewed by several groups [12,13,14]. Virtual screening identifies prospective molecular compounds using bioinformatics strategies that involve screening thousands of viable candidates to narrow down the hits that fit the functional and chemical desirability parameters. The observed enhanced accuracy in hits enables users to narrow down the screen for further evaluation using high-throughput screening, in vitro, and in vivo analysis [15,16].

The virtual screening method uses pre-built fragment libraries, such as PubChem, DrugBank, and ChEMBL [17,18,19,20]. Molecular docking is one such tool used to expand default libraries with fragments with structural complementarity to the target protein [21]. It involves two approaches. One is ligand-based algorithmic modeling, which utilizes a compound’s existing information and computationally imitates the binding capacities and biochemical outcomes of fragment–target interaction [22]. Another receptor-based computational method explores the molecular docking of each ligand into the target binding site, producing a predicted binding mode and a measure of the quality of the fit into the target binding site. The observed quality of fit determines the collected hits, which are then subjected to experimental testing for biological activity. This is exemplified in several other studies that identified natural compounds through library screening of marine invertebrate extracts that have been shown to have an anti-mycobacterial effect against Mtb [22,23,24].

There are other novel computational approaches, such as the linked-fragment approach [25], fragment-based automated site-directed drug design [26], dynamic ligand design [27], Caflisch’s technique [28], and the “core template” algorithm [29] for drug discovery. Purely computational approaches to drug discovery, including fragment design and discovery, are increasingly important, but are beyond the scope of this perspective. However, many of the applications discussed below rely on computational methods to prescreen fragments or to aid in their optimization and linkage.

### 2.2. X-ray Crystallography in Fragment-Based Drug Discovery

X-ray crystallography provides a map of the topographical structures of targeted chemical compounds and has been honed to improve the ‘positive hits’ during screening processes [16,30,31,32]. These results not only allow for the measurement of the ligand efficacy but also the binding precision of the particles [30]. Recent advancements in robotics, X-ray technology, and computation improvements in solving crystal structures have aided the discovery of several small-molecule inhibitors [32,33,34]. There have been two basic methods used for fragment screening using crystallography. The first method is crystal soaking, in which highly concentrated fragments diffuse through the crystal solvent channels and bind the target protein. Second, co-crystallization entails crystallizing both the fragment and the target protein [35]. Sherine et al. (2019) validated 27 of the 53 preliminary hits confirmed by differential scanning fluorimetry (DSF) using crystallography to target the tRNA-(N1G37) methyltransferase (TrmD) of *M. abscessus* [36]. In a recent study with X-ray crystallography data of DprE1 co-crystallized with the 2-carboxyquinoxaline series, the authors postulated the possibility of a structure-based analog design having anti-tuberculosis activity [7]. In addition, Frederickson et al. (2022) discovered high-affinity inhibitors and minimum inhibitory concentrations against Mtb using a combination of X-ray crystallographic and phenotypic screening methods [37]. In addition, Mahalingam et al. (2009) reported the use of crystallography in the early screening of heat shock protein 90 (HSP) inhibitors [38].

### 2.3. Biophysical Detection Methods: Thermal Shift and Mass Spectroscopy

Similar to the aforementioned screening procedures, biophysical detection methods rely on both the physical structure and biochemical outcome of ligand–receptor interactions. The general factors, such as pH value, the composition of the buffer, mutations in the amino acid sequence, or the binding of a ligand or fragment, affect the protein conformation, and ultimately its stability. The underlying idea of thermal shift analysis (TSA) is that the binding of a ligand alters the temperature at which a protein unfolds. Fluorescence-based TSA is common and usually referred to as thermofluor assays or differential scanning fluorimetry (DSF), in which the melting temperature of the protein can be determined. DSF involves protein being denatured by heat and the binding of fluorescent dye to the hydrophobic core of the protein, leading to an enhanced fluorescence signal [16]. Malapati et al. (2018) discovered MTB glutamate racemase inhibitor benzoxazole derivative 17 using DSF [39]. An additional putative inhibitor molecule, N-phenylphenoxy acetamide, which permeates the cell wall and membrane of *Mycobacterium smegmatis* (*M. smegmatis*) and inhibits transcriptional repressor EthR, was identified by DSF [7,40]. The first effective FBDD strategy utilized by Finzel and colleagues used DSF to screen new inhibitors of 7,8-Diaminopelargonic acid (DAPA) synthase (BioA) [41,42].

Another cutting-edge approach for fragment-based drug discovery is mass spectrometry (MS), which analyzes the mass-to-charge ratio of ions computationally. A mass analyzing component detects minute amounts (analytes) of a substance and separates the ions according to their charge and mass using a continuous ion source. This method of analysis provides information about the ions’ mass and structure, further identifying molecules that adhere to the target [43]. Swayze et al. (2002) successfully used the FBDD approach in combination with MS to extend the concept of “SAR by NMR” [5] to “SAR by MS” [44]. Extensive literature is available on the use of MS in the screening of novel inhibitors against Mtb CYP121 [45] and several other targets [43,46,47,48,49,50,51,52].

### 2.4. Cryo-Electron Microscopy

Cryo-electron microscopy (cryo-EM) has various advantages over more established techniques, such as X-ray crystallography or NMR, for achieving bigger molecular complexes (SBDD) [53]. Obtaining cryo-EM structures with better than 3 Å resolution is called the “resolution revolution” and has been proven to be a key milestone for cryo-EM [54,55]. In FBDD, advanced computational tools and cryo-EM have proven to be an effective combination. Zhou et al. published cryo-EM structures of cytochrome bcc from NTM and Mtb in combination with clinical drug candidates Q203 and TB47 in 2021. A study has demonstrated how Q203 and TB47 inhibitors work to prevent quinone binding to the Qo site, and hence prevent electron transfer [56].

### 2.5. Surface Plasmon Resonance

Surface plasmon resonance (SPR) is a frequently used potent method in fragment screening. SPR is employed in the screening of chemical libraries against a range of protein targets and is essential in determining the binding kinetics [57]. This technique can also be used to assess biomolecular interactions between different proteins, DNA/RNA, and small and complex molecules, such as binding affinity, specificity, and kinetic properties [16,35,58]. The determination of the refractive index during probe–target binding is a key step in SPR. In principle, the binding target protein is immobilized on the gold or silver surface and the protein solution is allowed to flow on the immobilized surface, which induces an increase in the refractive index upon probe–target binding [58,59,60]. SPR allows for the assessment of library sizes up to 100,000 fragments [61]. Earlier studies by Nikiforov et al., have successfully used fragment merging approaches based on SPR and other biophysical methods to develop small molecule inhibitors against Mtb EthR [62]. Recent reports have described the use of SPR in conjunction with other biophysical techniques to evaluate the ligandability of the ArgB, ArgC, ArgD, and ArgF in the Mtb L-arginine biosynthesis pathway [63]. Because the arginine biosynthesis pathway is conserved in *M. abscessus* and other NTMs, it opens up new possibilities for finding inhibitors of the d-novo arginine biosynthesis pathway.

### 2.6. Nuclear Magnetic Resonance

The ground-breaking strategy of “SAR by NMR” by Shuker et al. (1996) involves the use of NMR screening to characterize the structural properties of ligand–protein interactions and to identify the binding of small compounds to protein targets [5,16]. The key advantage of NMR screening is the determination of molecular-level interactions in a non-destructive manner along with the determination of dissociation constant values in the range of micromolar to millimolar [16,64]. Coupled with this technique, the field of fragment-based drug discovery has advanced in recent decades, granting biochemists and pharmacokinetic fields a greater understanding of ligand–receptor interactions. Based on a fragment screening strategy combining NMR and a biochemical assay, the researchers reported the discovery of 3-cyanopyridone and 1,6-naphthyridin-2-one as strong inhibitors of Mtb thymidylate kinase (TMK) [65]. Earlier studies have reported the potent inhibitor molecules against Mtb Pantothenate Synthetase using a series of biophysical techniques, including ligand-based NMR spectroscopy and NMR spectroscopy [66,67].

## 3. Applications

The NTM and Mtb pathophysiology combined with rising drug resistance represents the gravest danger to public health globally and necessitates the development of novel drug development approaches. However, for TB treatment, we have first-line drugs, such as isoniazid (INH), that kill 99.9% of Mtb cells, but there is a subpopulation that forms 0.1% of cells called persisters that become tolerant to drugs and probably lead to the emergence of drug resistance [68,69]. This persister population is not limited to Mtb but has been reported for many bacterial pathogens. The formation of biofilms by many mycobacteria further adds to the enhanced drug resistance. Finding targets that sterilize these persister subpopulations will aid in the inhibition of the emergence of drug resistance in Mtb, NTM’s, and other bacterial pathogens. Recently, in order to find potent drug inhibitors against potential targets in NTMs and Mtb, fragment-based techniques have proven to be significant and have substituted high-throughput screening [70]. The Mycobacteriaceae family comprises 174 known species, including the obligate pathogen Mtb and other NTMs, such as *M. abscessus* and *M. smegmatis*, etc. In addition to the structural complexity and other multitudes of factors, the complex genomic composition of these bacteria is key to developing drug resistance [71,72,73]. To date, a variety of compounds have been successfully investigated for their druggability and potential therapeutic efficacy against NTMs and Mtb targets (Table 1). The respective X-ray crystal structures and chemical structures of fragment hits are given in Table 2. The present section highlights target validation and efficacy studies where structure–activity relationships (SAR) have been successfully explored.

The enzyme Phosphoribosylaminoimidazole succinocarboxamide synthetase or PurC (also termed as SAICAR) found in *M. abscessus* is involved in de novo purine biosynthesis [92]. The synthesis of phosphoribosylaminoimidazole-succinocarboxamide (SAICAR) from 5-aminoimidazole-4-carboxyribonucleotide (CAIR) and l-aspartate in the presence of adenosine triphosphate (ATP) and Mg2+ cofactors is a key step for this enzyme [93,94]. The non-homologous nature of the enzyme *M. abscessus* SAICAR synthetase with human ortholog makes it an ideal candidate for drug inhibition studies [70]. Charoensutthivarakul et al. (2022) are the first group to explore the importance of PurC in *M. abscessus* and the identification and validation of the potential inhibitor fragments against the SAICAR. Followed by SAR studies, hit-to-lead optimization, and phenotypic screening of compounds, the authors have reported promising inhibitory activity against *M. abscessus* and Mtb [11].

Tiwari et al. (2018) identified the upregulation of the de novo L-arginine biosynthesis pathway as an early adaptive response to the oxidative stress generated by vitamin C and INH [95,96]. Arginine deprivation of Mtb quickly sterilizes Mtb persister populations, leading to in vitro and in vivo sterilization of MtbΔ*argB* and Δ*argF* mutants without the appearance of suppressor mutants. Recently, our lab has used a fragment-based approach to assess the ligandability of ArgB, ArgC, ArgD, and ArgF enzymes involved in L-arginine biosynthesis in the biosynthetic pathway. Following the identification of several hits against these enzymes and validation with biochemical and biophysical assays as well as X-ray crystallographic studies, compounds with on-target activity against ArgB of Mtb were confirmed [63,96]. Because the arginine biosynthesis pathway is conserved in *M. *abscessus** and other NTMs, it opens up new possibilities for finding inhibitors of the de novo arginine biosynthesis pathway [97].

For fragment-based whole cell screening, Moreira et al. (2016) used the poly-pharmacology, physicochemical, and pharmacokinetic properties, such as multi-target activity, small size, and hydrophilic nature, of the first-line TB drug “pyrazinamide” [98,99,100]. After further screening of 1725 fragments at a single concentration against *M. bovis* BCG, they found 116 primary hits that were selected for growth inhibitory activity. Out of these, 38 were found to be active against Mtb and one-third of these compounds have shown inhibitory activity against *M. abscessus* and *M. avium* [100].

Methylation is another intricate and unexplored mechanism involved in survival and immune evasion by Mtb in the host. The genome of Mtb encodes 121 methyltransferases (MTases). The connection of MTases with critical cellular events enables the researcher to design novel anti-tuberculosis treatment strategies [101]. The tRNA-(N^1^G37) methyltransferase (TrmD) is one such enzyme which catalyzes the transfer of a methyl group from AdoMet to the N^1^ position of the G37 base required for the synthesis of m^1^G37 on tRNA [102]. This process of methylation is necessary to suppress tRNA frameshifting during protein synthesis [103]. Transfer RNA (tRNA) modifications play a crucial role in bacterial pathogenesis. A study by Chionh et al. (2016) reported *M. bovis* BCG survival in hypoxic conditions with the help of a selective translation of codon-biased persistence genes mRNAs [104,105]. In *E. coli*, tRNA (m^1^G37) methyltransferase (TrmD) participates in reading frame maintenance and frameshift error prevention during transcription [106]. The researchers used a fragment-based approach to merge two fragments bound to the TrmD active site and, followed by employing guided elaboration, successfully designed potent nanomolar inhibitors. They demonstrated promising MIC values in in vitro testing against *M. abscessus* and Mtb [36,74]. Recently, researchers investigated the role of Mtb Rv1523 methyltransferase in methylation-mediated cell wall remodeling and modulation of host immune responses, imparting virulence and drug resistance [107].

Earlier studies have shown that Thymidylate kinase (TMK) has an essential role in DNA biosynthesis. Several biochemical and biophysical studies have reported that the thymidylate kinase of Mtb is an attractive target for structure-based drug design in the search for multidrug-resistant TB [96,97]. The study by Naik et al. (2016) reported that two novel classes of inhibitors, 3-cyanopyridones and 1,6-naphthyridin-2-ones, are involved in the synthesis and in vitro cellular activity against Mtb. Interestingly, the inhibitor (6-naphthyridin-2-one) concentration could be reduced to 200 nM (from 500 M) by fragment hit, with increased ligand effectiveness against TMK of Mtb [65]. A recent study of genomic comparison between *M. abscessus* subspecies has revealed the microevolutionary differences among the three subspecies based on the presence of TMK, 30S ribosomal protein S3, and 46 other proteins [108,109]. This signifies the possible involvement of TMK in drug discovery against NTMs.

Aspartate decarboxylase PanD is another enzyme that plays an essential role in the synthesis of cofactor coenzyme A in Mtb and is targeted by the first-line anti-TB drug pyrazinamide. The Mtb amidase converts pyrazinamide into its bioactive form, pyrazinoic acid (POA). Ragunathan et al. (2021) identified POA-based inhibitors of Mtb PanD with improved inhibitory activity against the enzyme Mtb PanD based on structure–function analyses [91]. In a recent study, Saw et al. (2022) identified (3-(1-naphthamido)pyrazine-2-carboxylic acid) (analogue 2) as having potent inhibitory activity against *M. abscessus* PanD. Additionally, analogue 2 showed whole-cell activity against three subspecies of the *M. abscessus* complex [75].

The pantothenate synthetase (PS) gene in Mtb encodes an important target in tuberculosis treatment. PS generates pantothenate in an ATP-dependent manner by forming an amide bond between pantoate and alanine [110,111]. The indispensable nature of pantothenate biosynthesis and the non-homologous nature of pantothenate synthetase for humans suggest that the enzyme involved in pantothenate biosynthesis could be a crucial target for anti-TB drug development. Jacobs and co-authors reported the importance of the pantothenate biosynthetic pathway in Mtb virulence [112]. Wang and Eisenberg elucidated the crystal structure of PS and showed that a tetrahedral intermediate generated from the β-alanine and pantoyl adenylate reaction can serve as an excellent target for inhibitor design [110]. Group efficiency analysis including both fragments’ growing and linking approaches was successfully used in the discovery of potent inhibitors against Mtb pantothenate synthetase. Furthermore, when tested in vitro, it had on-target inhibitory action against Mtb [113].

Phosphopantetheine Adenylyltransferase (PPAT) is another important enzyme in the biosynthesis of coenzyme A. The ATP-dependent conversion of 4′-phosphopantetheine adenylylation to 3′-dephospho-CoA and pyrophosphate is catalyzed by PPAT. The steps for CoA biosynthesis pathways are universally conserved in all species, including pathogenic microorganisms, with a significant difference in sequence, structure, and mechanistic level [114]. However, the structural difference between bacterial and human PPATs highlights the possible potential of the enzyme to be a novel drug target [71,114]. Thomas et al. (2022) used the FBDD approach to structurally characterize PPAT and validate the ligandability of 4-Amino-6-(pyrazol-4-yl)pyrimidine derivatives against *M*. *abscessus* [115]. This characterization was based on the DSF for fragment screening and X-ray crystallography for hit validation. Additionally, authors have identified series of compounds having in vitro activity against the Mtb PPAT and tested these identified compounds against *M.*
*abscessus.* They discovered that one of the compounds (compound **20**) has a low molecular affinity for M. abscessus PPAT. However, when tested for the whole-cell model, it failed to show significant inhibitory activity [71]. A study by Bakali et al. (2020) used a three-stage biophysical screening cascade-based FBDD approach to screen the library of 1265 compounds. The three chemically distinct fragment hits (benzophenone 1, indole 2, and pyrazole 3) were discovered in the screen, as well as additional low-micromolar non-natural Mtb PPAT active site binders with proven on-target in vitro whole-cell activity, as demonstrated using CRISPRi [116]. Additionally, Primi et al. (2021) screened 78 compounds followed by ligand-based drug design (LBDD) and SAR strategy for structural optimization against Mtb PPAT. They found the optimum activity of these compounds in anti-Mtb and MtbPPAT inhibition assays, with the highest activity of the MCP163 compound in both assays [117].

MmpL3 is an inner membrane mycolic acid transporter found on the cell envelope of all mycobacteria [118]. Dupont et al. (2016) identified a new piperidinol-based molecule, PIPD1, which targets MmpL3 when tested against *M. abscessus* and Mtb [97,119,120]. Recently, based on molecular modeling, SAR, and in vivo toxicology and pharmacokinetics studies, Ruyck et al. (2020) synthesized a series of piperidinol derivatives. When tested for their biological activity against *M. abscessus*, the authors found a new promising and valid analogue, namely FMD-88, with good pharmacokinetic properties, such as organ distribution after IP administration and elimination with a half-life of about 3 h. These results highlight the importance of anti-MmpL3 inhibitors as a weapon against *M. abscessus* [120].

Interestingly, multidrug tolerance, biofilm formation, and persistence under stressed conditions in Mycobacteria have been associated with the toxin-antitoxin (TA) systems [121]. Naturally, the involvement of TA systems in these important regulatory pathways makes them natural targets for drug discovery [121,122]. In recent research by Kim et al. (2022), the authors have prepared an antitoxin of type 2 TA class molecule (Mab3862) using molecular cloning from *M. abscessus.* Using in silico guidance and the FBDD approach, the authors have discovered a few valid hits that bind to novel sites in the de novo structure of Mab3862 antitoxin [76].

Additionally, in search of novel potential Mtb inhibitors, the FBDD approach has successfully explored several other molecules, such as Cytochrome P450 enzymes [45,123], malate synthase-G [124,125,126], CYP121 [45,123], DNA gyrase [127,128,129], and Inosine-5′-monophosphate Dehydrogenase [88,130,131,132,133,134].

Earlier, several studies have reported considerable sequence similarities between *M. abscessus* and Mtb [77,135,136]. DNA sequence similarity predicts that *M. abscessus* and Mtb share many biochemical pathways. This suggests that the already available TB regime could be used as a starting point to “jump start” *M. abscessus* drug discovery projects [77,137]. Dupont et al. (2016) have successfully used this approach for the synthesis of the new piperidinol-based compound from the already established tubercular hits [119], which shows potential inhibitory activity against *M. abscessus* [97]. Therefore, based on the findings and observations from earlier studies, we suggest that the data collected during earlier TB research studies can be used as a foundation for the development of novel therapeutic molecules against NTMs.

## 4. Challenges in Fragment-Based Drug Discovery

Drug discovery for NTMs and MTB has been successfully accomplished using the FBDD. This strategy, however, faces a number of obstacles and challenges with its methodologies. We have covered a lot of them in this area, which will help to overcome the challenges listed below and pave the way for successful drug discovery (Table 3).

### 4.1. Specialized Methods Are Needed to Detect Fragment Binding in Libraries

The fragments form high-attribute interactions if they have suitable ligand efficiencies (LE), which are ineffective due to their smaller size and weak affinity [138]. It is well known that selected fragment libraries follow the “rule of three”. For example, using three-dimensional fragments instead of flat structures, identifying differences at fragment binding sites and protein binding sites, and determining the potent target hits for fragment protein interaction. Although there are arguments against using the rule of three, it is considered the preferred model for fragment selection in libraries [16,139,140]. Various approaches, such as NMR, surface plasmon resonance (SPR), and X-ray screening, have greater sensitivity and are specifically used to screen larger fragments in higher libraries in fragment-based drug discovery (FBDD) [30]. Additionally, thermal denaturation, thermal electrophoresis, capillary electrophoresis, mass spectrometry, and isothermal titration calorimetry have also been widely used in FBDD despite their limitations.

### 4.2. Optimization of Fragment Hits Using Computational Tools

Computational tools, such as de novo design, combinatorial docking, and interactive optimization, and their approaches, play a key role in the optimization process through fragment linking that eventually reduces the time required for increasing the efficacy of potent fragment leads [141,142]. Optimization and proper knowledge of specific fragments is a pre-requisite for FBDD [143]. The large size of targeted molecules or the existence of targets as multiprotein complexes significantly impacts the generation of robust crystals, leading to the hindrance of using the FBDD approach for screening fragments [8]. The optimization of fragment hits also depends on detection techniques considered for FBDD; the correlation of fragment hits must be carefully considered. For example, 5% of fragment interaction of a protein is sufficient for an NMR hit, whereas in X-ray crystallography it requires 30–70% binding to be considered as a potential hit [144,145,146]. Studies have also proposed that the quality of binding hits obtained from crystallography is more rigid and qualitative compared to techniques such as SPR, mass spectrometry, calorimetry, and NMR [147].

### 4.3. Modeling

During modeling procedures, smaller fragments, compared to the target enzyme, make molecular docking more difficult due to a lack of interactions with the residual molecules and low-affinity binding to the functional groups [8]. Under these conditions, the fragment analysis might turn out to be difficult as some potent fragment hits may have been missed due to weaker interactions and the smaller size of the protein. Thus, the smaller size of the fragments might not be promising candidates for docking unless they share related physical-chemical properties and binding pockets that eventually lead to the time-consuming subsequent analysis [148]. Among the modeling procedures chosen for FBDD, docking has advanced in recent years. Researchers have various options that can be chosen either from stoichiometric methods by following empirical or conformational changes, force-field, or using a systematic approach [149]. However, even though all of these factors were considered, the accuracy remains unclear due to the receptor–ligand binding efficacy. To avoid these conditions, standard protein models and potent binding sites are essential for FBDD. The majority of the membrane proteins, such as G protein-coupled receptors (GPCRs), do not have any information on crystallization, which is challenging in the development of new drugs [150]. Homology modeling has answered its insights into this problem by targeting proteins with homology sequences and crystallizing the targeted proteins using static models [151,152]. Homology modeling is dependent on changes in binding site conformation and the shape of the protein structure. Despite these drawbacks, artificial intelligence algorithms and new chemical informatics were also improved for in silico drug design using molecular dynamic simulations [145,153]. With this approach, potent drug candidates can be assessed early during fragment-based drug discovery.

### 4.4. Challenges in Hit Identification and Lead Optimization

Proteins encoded by essential genes play a crucial role in the survival of organisms. Several studies after the publication of the first Mtb genome sequence have determined essential genes and, using a target-based approach, they have designed novel inhibitors against them [7]. Understanding the molecular targets and modes of action is necessary to develop a drug for the treatment of tuberculosis without failure. In recent times, Mtb therapeutics has been focusing on identifying effective compounds that specifically inhibit the survival of the bacteria and proliferation in the host. The genome of Mtb encodes several virulent proteins that are secreted to alter the host’s immune system, which helps to improve resistance against therapeutics [154,155]. Therefore, it limits the pathogen’s activities in the host by inhibiting the essential pathogenic proteins. The key feature of the fragment-based approach is to isolate the protein in a larger quantity and maintain its stability for the screenings. Further, crystallization of the protein target is required to determine the binding modes. The fragment-based approach is difficult without stability and high yields of protein. Additionally, genetic screen trials are the first steps in identifying the genetic products that will be the focus of chemotherapy against Mtb. However, not all important genes are similarly responsive to pharmacological treatment [156]. The lower molecular weight compounds with inhibitor action must have specificity to affect the target function without the interference of any host orthologs [157,158].

The fragment-based approach is now an established and effective method to develop compounds that can inhibit or activate the function of a target protein. The small molecules (<18 heavy atoms) exhibit their functionality to bind and are small enough to fit into the active site of the protein, which is a major advantage over larger compounds that block the binding site of the protein molecule [159]. Fragment screening in the library is the first step to screening small molecules using various biophysical and functional assays. Another important screening approach is soaking the fragments with a known crystal structure of the protein, which is useful to detect weak binding and characterize the binding position of the fragment [35]. However, it has been challenging to prepare larger quantities of soaked crystals, collect the data in higher amounts, and complete the processing necessary to investigate the thousands of diffraction datasets to screen every single compound in the fragment library.

The screening of library fragments against an enzyme generates a significant number of hits. The absolute hit rate depends on the target, but usually most fragment screens result from multiple hits are more than 10, which can be developed into a different chemical series. The fragment growth strategy, used for developing the fragment hits, is effective when targeting the enclosed active site pockets and the effect of each functional group or atom is to be assessed thoroughly; hence, maintaining optimal ligand efficiency. It is preferable that the target protein should have compact sites that are capable of ligand-efficient interactions. In contrast, fragment linking might be responsible for a less ligand-efficient site that has multiple subpockets. For example, the potency fragments can be developed into potent inhibitors. For example, pantothenate synthase of Mtb, which is the logically direct way [66]. During each synthetic cycle, the analysis of enzyme–ligand complexes and ligand efficiencies is useful to guide the rational design. Therefore, there is no chance of incorrect assumptions during the optimization process. Similarly, the detailed structural analysis helps to guide the selection of linkers and in the understanding of the imposed conformational restraints. Although it appears more elegant, the repertoire of linkers is limited and compromises the binding of the original. On the other hand, flexibility will be present at each stage in the fragment-growing approach and will allow more space for further optimization.

### 4.5. Post-Modeling Expression, Solubilization, Purification, Crystallization, Data Collection, and Structure Solution

X-ray crystallography is an applicable technique to validate any hits during the screening process. The crystal structure reveals the binding mode of the fragment. However, crystallography requires bulk expression and a high amount of pure protein in a soluble form, which is challenging and time-consuming. Specifically, membrane proteins are difficult to crystallize due to the presence of hydrophobic transmembrane regions (TMRs). Crystal soaking is preferable as it leads to the high-affinity interactions. However, it needs further investigation. When the fragment is not soluble, then co-crystallization can be used, and only one fragment is crystallized with the target protein [160]. Crystal formation, data collection, and analysis are limiting factors for the FBDD approach, but robotic automation has made the overall crystallization process simpler and more convenient as it has made it feasible to work with smaller quantities of proteins.

The new generation of detectors is the fastest data collection tool. For example, they have termed pixel array detectors (PADs) an emerging technology and synchrotron beamlines widespread. Structures can be solved by molecular replacement using PHASER [161] and refinements can be achieved using REFMAC [162] and PHENIX [163]. Many programs are available to build ligands into electron densities. Commercial packages include PrimeX by Schrödinger (New York, NY, U.S.A.), Rhofit by Global Phasing (Cambridge, U.K.), and Afitt by OpenEye (Santa Fe, NM, U.S.A.) [164]. However, the major pitfalls of X-ray crystal structures are fragment screening and lead design, which depend on the skill of the crystallographer regardless of resolution. Uncertainty in individual atom positions and amino acid placements, changes in solvent structures, and solvent binding interpretation become more difficult.

## 5. Conclusions and Future Perspectives

Non-tuberculous mycobacteria (NTMs), which commonly inhabit the environment, can cause disease in immunocompromised as well as immunocompetent individuals. NTM infections are often incredibly difficult to treat due to intrinsic drug resistance and biofilm formation. They require prolonged multi-drug therapy, which has also been linked with the increased emergence of antibiotic resistance. The most common NTM strains associated with pulmonary infections are members of the *Mycobacterium abscessus* complex (MABC). Despite the effectiveness of first-line treatments for TB, the risk of arising persister subpopulations can be the cause of emerging drug resistance, which is not only limited to Mtb but also to bacterial pathogens, such as NTM’s. FBDD has been a proven effective approach for TB drug discovery because of its straightforward and target-driven strategy. Recent studies using the FBDD approach have been conducted in search of novel inhibitors against NTMs and Mtb targets with proven potent in vitro and in vivo activity, with some of them currently in clinical development. Additionally, overcoming the mentioned difficulties observed during FBDD, such as modeling and post-modeling challenges, will pave the way for the removal of hindrances in the way of drug discovery. In addition, the review highlights the importance of the combined approach of the several domains, technological aspects, and the challenges in FBDD against anti-NTM and anti-TB drug discovery.

We believe that given the present status of drug molecules in the clinical trial stage, continuous and enormous technological advancement in the FBDD field accelerates and makes efficient the entire process of drug discovery against mycobacteria.

## Figures and Tables

**Figure 1 ijms-23-10669-f001:**
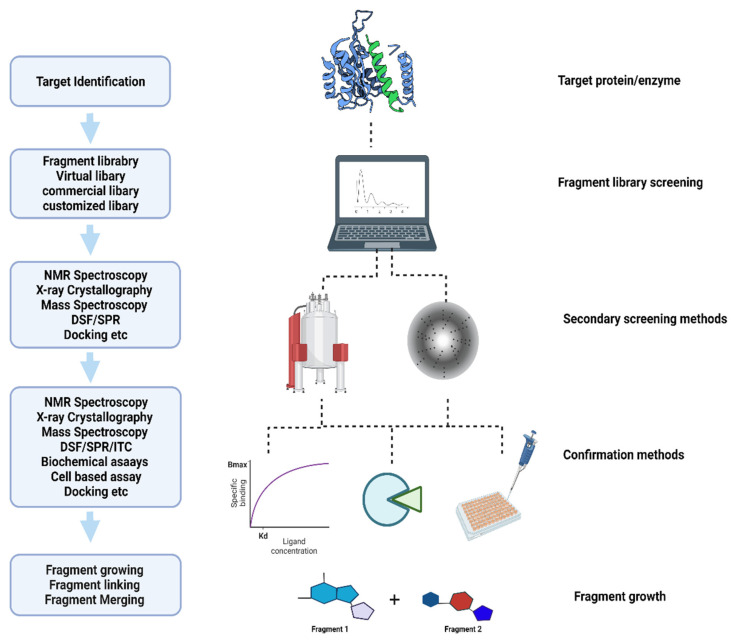
Flow chart of fragment-based drug discovery.

**Table 1 ijms-23-10669-t001:** Tested analogues, target compounds, and their status.

Target	Analogue	Status	Reference
SAICAR (PurC)	4-Amino-6-(pyrazol-4-yl)pyrimidine Derivatives	In vitro activity	[11]
ArgB	NMR711, NMR446, and L-canavanine	In vitro activity	[63]
tRNA (m^1^G37) methyltransferase	Series of compounds	In vitro activity	[74]
Thymidylate Kinase	4-[3-cyano-2-oxo-7-(1H-pyrazol-4-yl)-5,6-dihydro-1H-benzo[h]quinolin-4-yl] ben zoic acid	In vivo activity	[65]
Aspartate decarboxylase (PanD)	(3-(1-naphthamido)pyrazine-2-carboxylic acid) (analogue 2)	In vivo activity	[75]
type 2 TA class	Mab3862	In vitro activity	[76]
MmpL3	PIPD1	In vivo activity	[77]
Decaprenylphosphoryl-β-D-ribofuranose 2′-oxidase	Benzothiazinone, Macozinone	Phase 1 clinical trial	[78,79]
Mycobacterial membrane protein Large 3	SQ109	In vivo limited activity	[80,81,82]
Enoyl Reductase (InhA)	Tetrahydrobenzothieno pyrimidines	In vitro activity	[7]
Enoyl Reductase (InhA)	Thiadiazolyl methylthiazoles	In vivo limited activity	[7,83]
β-ketoacyl ACP synthase I (KasA)	Indazole sulfonamides (GSK3011724A)	In vitro activity	[84]
Cytochrome P450 (CYP121)	4-(3-amino-1H-pyrazol-4-yl)phenol	In vitro activity	[45]
type-2 NADH dehydrogenases	quinolinyl pyrimidine series	In vitro activity	[85]
type-2 NADH dehydrogenases	2−Mercapto quinazolinones	In vitro activity	[86,87]
EthR transcriptional repressor	N-phenylphenoxy acetamides	In vitro activity	[7]
Inosine-5′-monophosphate dehydrogenase	Indazole sulfonamides	In vitro activity	[88]

**Table 2 ijms-23-10669-t002:** X-ray crystal structures and respective fragment hit chemical structures.

Structure/X-ray Crystal Structure	Fragment Hit	References
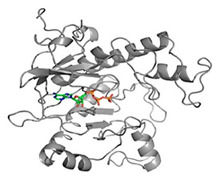	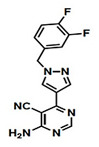	[11]
SAICAR Synthetase (PurC)	4-Amino-6-(pyrazol-4-yl)pyrimidine Derivatives	
* 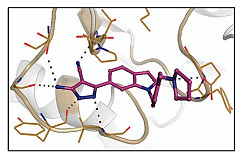 *	* 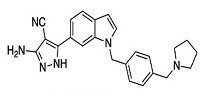 *	[36,74]
tRNA (m1G37) methyltransferase	3-Amino-5-(1-(4-(piperidin-1-ylmethyl)benzyl)-1H-indol-6-yl)-1H-pyrazole-4-carbonitrile derivatives	
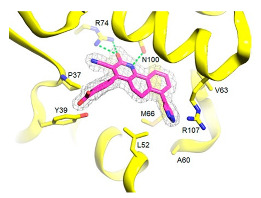	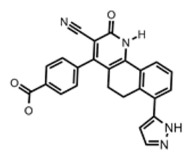	[65]
Thymidylate Kinase	4-[3-cyano-2-oxo-7-(1H-pyrazol-4-yl)-5,6-dihydro-1H-benzo[h]quinolin-4-yl] benzoic acid	
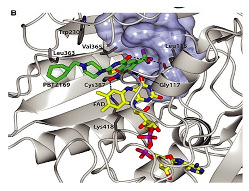	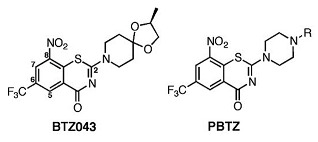	[79]
DprE1-PBTZ169 complex	Benzothiazinone, Macozinone	
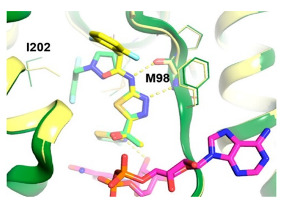	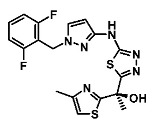	[83,89]
Enoyl Reductase (InhA)	Methylthiazoles	
* 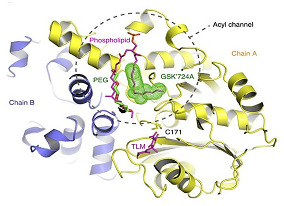 *	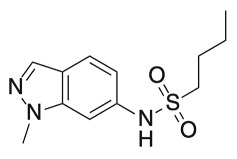	[84]
β-ketoacyl ACP synthase I (KasA)	Indazole sulfonamides (GSK3011724A)	
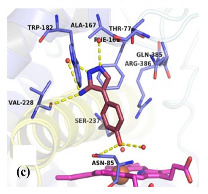	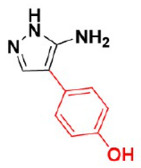	[45,90]
CYP121 in complex with retrofragments 4	4-(3-amino-1H-pyrazol-4-yl)phenol	
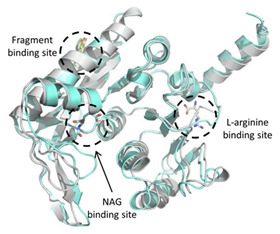	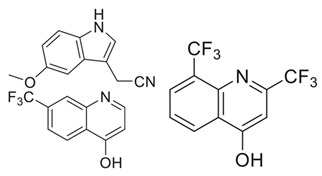	[63]
Acetyl Glutamate Kinase (ArgB)	NMR711 and NMR446	
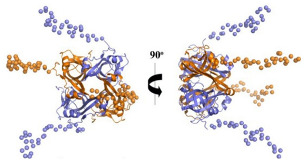	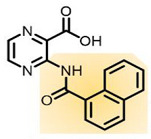	[75,91]
Coral model of Mab Aspartate decarboxylase (PanD)	(3-(1-naphthamido)pyrazine-2-carboxylic acid) (analogue 2)	

**Table 3 ijms-23-10669-t003:** Challenges in the FBDD approach.

Method/Molecules	Challenges
X-ray crystallography	− Bulk expression and solubility − Crystal formation, data collection and analysis
X-ray crystal structures	− Fragment screening and lead design − Positions of the atom, amino acid placements, changes in solvent structures, and solvent binding interpretations
Crystal soaking	− Further investigation for high-affinity interactions − Prepare larger quantities of soaked crystals
Molecular docking with smaller fragments	− Low affinity binding − Lack of interactions − Fragment analysis
Homology modeling	− Conformational changes in binding sites and shape of the protein structure
Stoichiometric approach	− Accuracy of receptor–ligand binding efficiency
Fragment screening	− Large size or target and multiprotein complexes
Fragments	− Smaller size and weak affinity − Flat structured fragments
Low efficiency fragments	− Optimization
Fragment potential hits	− Quality of binding

## Data Availability

Not applicable.

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
