# Peer review of "Fragment-Based Drug Discovery against Mycobacteria: The Success and Challenges"

_ijms, 2022, doi:10.3390/ijms231810669_

Round 1

Reviewer 1 Report

This review article provides the basics of fragment based drug discovery and goes into the recent successful utilization of (FBDD) in identifying targets and developing inhibitors against Mycobacteria. Overall, the review is well structured and well written. I do not have any comments on improving the manuscript, but there are several spelling errors in Figure 1.

Customized is spelled wrong

Fragment is spelled wrong.

Please rectify those. 

Author Response

This review article provides the basics of fragment based drug discovery and goes into the recent successful utilization of (FBDD) in identifying targets and developing inhibitors against Mycobacteria. Overall, the review is well structured and well written. I do not have any comments on improving the manuscript, but there are several spelling errors in Figure 1.

Customized is spelled wrong

Fragment is spelled wrong.

Please rectify those.

Author Response: We thank reviewer for their suggestions. We have corrected the mentioned spelling errors “Customized” as well as “Fragment” in Figure 1.

Reviewer 2 Report

This review is describign about fragment-based drug discovery against Mycobacteria, specially to the success and challenges. This review focuses on  the recognition of fragment-based drug discovery as a popular approach and on challenges associated with the FBDD approach in identification and development of novel compounds. It is recommended to be acceptable after minor revisions.

- For Section 4,  a Table can help the clear understading for readers. Please make a Table.

- English should be checked thoroughly. (ex, L10, pose -> poses)

Author Response

This review is describing about fragment-based drug discovery against Mycobacteria, specially to the success and challenges. This review focuses on the recognition of fragment-based drug discovery as a popular approach and on challenges associated with the FBDD approach in identification and development of novel compounds. It is recommended to be acceptable after minor revisions.

- For Section 4,  a Table can help the clear understanding for readers. Please make a Table.

- English should be checked thoroughly. (ex, L10, pose -> poses)

Author Response: We thank reviewer for their suggestions. We have added Table 3 for Section 4 as mentioned by the reviewer for the clear understanding for the readers. We have also checked manuscript for the grammatical corrections.

Reviewer 3 Report

The review by Togre et al. offers a very complete perspective on drug discovery strategies, applications, and challenges using the fragment-based drug discovery approach. The topic is relevant because tuberculosis is related to high rates of morbidity and mortality worldwide. In addition, a new facet is the occurrence of multidrug-resistant strains, which represents a major challenge for effective treatment. Another issue is that there are few recent reviews on the subject. In fact, the last one I found with the same focus was published in 2017. The manuscript submitted to IJMS presents more interesting information about the strategies that have been used on fragment-based drug discovery to date. However, a revision of this nature requires some additional figures to the one inserted. I suggest the following to make it more attractive and didactic:

- X-ray crystal structures

- chemical structures of fragment hits

The conclusions are appropriate, that is, they are consistent with the data presented and address the main question.

In general, the content is deep and well-written, but the text needs a moderate review of style and grammar. I strongly recommend it for publication after a minor revision.

Author Response

The review by Togre et al. offers a very complete perspective on drug discovery strategies, applications, and challenges using the fragment-based drug discovery approach. The topic is relevant because tuberculosis is related to high rates of morbidity and mortality worldwide. In addition, a new facet is the occurrence of multidrug-resistant strains, which represents a major challenge for effective treatment. Another issue is that there are few recent reviews on the subject. In fact, the last one I found with the same focus was published in 2017. The manuscript submitted to IJMS presents more interesting information about the strategies that have been used on fragment-based drug discovery to date. However, a revision of this nature requires some additional figures to the one inserted. I suggest the following to make it more attractive and didactic:

- X-ray crystal structures

- chemical structures of fragment hits

The conclusions are appropriate, that is, they are consistent with the data presented and address the main question.

In general, the content is deep and well-written, but the text needs a moderate review of style and grammar. I strongly recommend it for publication after a minor revision.

Author Response: We thank reviewer for their suggestions and we agree with the reviewer that addition of X-ray crystal structures and the chemical structures of their respective hits makes the manuscript impactful. Therefore, we have added Table 2. We have also checked manuscript for the grammatical corrections